# MANIFOLD ALIGNMENT VIA FEATURE CORRESPONDENCE

## ABSTRACT

We propose a novel framework for combining datasets via alignment of their associated intrinsic dimensions. Our approach assumes that the two datasets are sampled from a common latent space, i.e., they measure equivalent systems. Thus, we expect there to exist a natural (albeit unknown) alignment of the data manifolds associated with the intrinsic geometry of these datasets, which are perturbed by measurement artifacts in the sampling process. Importantly, we do not assume any individual correspondence (partial or complete) between data points. Instead, we rely on our assumption that a subset of data features have correspondence across datasets. We leverage this assumption to estimate relations between intrinsic manifold dimensions, which are given by diffusion map coordinates over each of the datasets. We compute a correlation matrix between diffusion coordinates of the datasets by considering graph (or manifold) Fourier coefficients of corresponding data features. We then orthogonalize this correlation matrix to form an isometric transformation between the diffusion maps of the datasets. Finally, we apply this transformation to the diffusion coordinates and construct a unified diffusion geometry of the datasets together. We show that this approach successfully corrects misalignment artifacts, and allows for integrated data.

## 1 INTRODUCTION

In biology and other natural science settings we often have the problem that data are measured from the same system but with different sensors or in different days where sensors are calibrated differently. This is often termed *batch effect* in biology and can include, for example, drastic variations between subjects, experimental settings, or even times of day when an experiment is conducted. In such settings it is important to globally and locally align the datasets such that they can be combined for effective further analysis. Otherwise, measurement artifacts may dominate downstream analysis. For instance, clustering the data will group samples by measurement time or sensor used rather than by biological or meaningful differences between datapoints.

Recent works regard the two datasets as views of the same system and construct a multiview diffusion geometry but all of them require at least partial bijection, if not full one, between views (Lafon et al., 2006; Ham et al., 2003; 2005; Wang & Mahadevan, 2008; Tuia & Camps-Valls, 2016). Other work directly attempt to match data points directly in ambient space, or by local data geometry and these can be very sensitive to differences in sampling density rather than data geometry (Haghverdi et al., 2018). Here, we present a principled approach called *harmonic alignment* to for correct this type of effect based on the manifold assumption.

The manifold assumption holds that high dimensional data originates from an intrinsically low dimensional smoothly varying space that is mapped via nonlinear functions to observable high dimensional measurements. Thus, we assume that the datasets are from transformed versions of the same low dimensional manifold. We learn the manifolds separately from the two datasets using diffusion geometric approaches and then find an isometric transformation to map from one manifold to the other. Note that we are not aligning points to points. Indeed there may be sampling differences and density differences in the data. However, our manifold learning approach uses an anisotropic kernel that detects the geometry of the data to align rather than point-by-point matching which is done in other methods.

Our method involves first embedding each dataset separately into diffusion components, and then finding an isometric transformation that aligns these diffusion representations. To find such transformation, we utilize the duality between diffusion coordinates and geometric harmonics that act as generalized Fourier harmonics in the graph space. The diffusion components are eigenvectors of a Markov-normalized data diffusion operator, whose eigenvalues indicate frequency of the eigenvector. We attempt to find a transformation from one set of eigenvectors to another, via feature correspondences in the data.

While datapoint correspondences may be difficult or impossible to obtain since many biological measurements are destructive, feature correspondences are often available. For instance single-cell measurements of cells from the same device, thus containing counts for the same genes, albeit affected by batch differences. Thus when corresponding features are transformed via the graph Fourier transform (GFT) into diffusion coordinates, the representations should be similar, with potentially small frequency-proximal perturbations. For instance, slowly varying features across the manifold should be load to low-frequency eigenvectors of the Markov matrix. This insight allows us to create a correlation matrix between the eigenvectors of one dataset to another based on correlation between feature loadings to the eigenvectors. However, since we know that eigenvectors represent frequency harmonics, we need not compute the entire correlation matrix but rather only the near-diagonal values. This implies that the two manifolds must be perturbed such that low and high frequency eigenvector space are similar. We then find a linear transformation between that maps the eigenvectors of one space into those of the other that maximizes these correlations by orthogonalizing this matrix. This transformation allows us to align the two datasets with each other. Finally, given an aligned representation, we build a robust unified diffusion geometry that is invariant batch effects and sample-specific artifacts and low-pass filter this geometry to denoise the unified manifold. Thus, in addition to aligning the manifolds our method denoises the manifolds as well.

We demonstrate the results of our method on artificial manifolds created from rotated MNIST digits, corrupted MNIST digits, as well as on single-cell biological data measuring peripheral blood cells. In each case our method successfully aligns the manifolds such that they have appropriate neighbors within and across the two datasets. We show an application to transfer learning where a lazy classifier trained on one dataset is applied to the other dataset after alignment. Further, comparisons with recently developed methods such as the MNN-based method of Haghverdi et al. (2018) show significant improvements in performance and denoising ability.

## 2 HARMONIC ALIGNMENT

A typical and effective assumption in machine learning is that high dimensional data originates from an intrinsic low dimensional manifold that is mapped via nonlinear functions to observable high dimensional measurements; this is commonly referred to as the manifold assumption. Formally, let $\mathcal{M}^d$ be a hidden $d$ dimensional manifold that is only observable via a collection of $n \gg d$ nonlinear functions $f_1, \ldots, f_n : \mathcal{M}^d \to \mathbb{R}$ that enable its immersion in a high dimensional ambient space as $F(\mathcal{M}) = \{\mathbf{f}(z) = (f_1(z), \ldots, f_n(z))^T : z \in \mathcal{M}^d\} \subseteq \mathbb{R}^n$ from which data is collected. Conversely, given a dataset $X = \{x_1, \ldots, x_N\} \subset \mathbb{R}^n$ of high dimensional observations, manifold learning methods assume its data points originate from a sampling $Z = \{z_i\}_{i=1}^N \in \mathcal{M}^d$ of the underlying manifold via $x_i = \mathbf{f}(z_i)$, $i = 1, \ldots, n$, and aim to learn a low dimensional intrinsic representation that approximates the manifold geometry of $\mathcal{M}^d$.

A popular approach towards this manifold learning task is to construct diffusion geometry from data (Coifman & Lafon, 2006), and embed data into diffusion coordinates, which provide a natural global coordinate system derived from Laplace operator over manifold geometries, as explained in Section 2.1. However, this approach, as well as other manifold learning ones, implicitly assumes that the feature functions $\{f_j\}_{j=1}^n$ represent data collection technologies (e.g., sensors or markers) that operate in a consistent manner on all samples in $Z$. While this assumption may be valid in some fields, biological data collection (and more generally, data collection in empirical sciences) is often highly affected by a family of phenomena known as *batch effects*, which introduce nonnegligible variance between different data batches due to various uncontrollable factors. These include, for example, drastic variations between subjects, experimental settings, or even times of day when an experiment is conducted.

Therefore, in such settings, one should consider a collection of $S$ samples $\{X^{(s)}\}_{s=1}^{S}$, each originating from feature functions $\{f_j^{(s)}\}_{j=1}^{n}$ that aim to measure the same quantities in the data, but are also affected by sample-dependent artifacts. While each sample can be analyzed to find its intrinsic structure, their union into a single dataset $X = \bigcup_{s=1}^{S} X^{(s)}$ often yields an incoherent geometry biased by batch effects, where neither the relations between samples or within each sample can be clearly seen. To address such artifacts, and constrct a unified geometry of multiple batches (i.e., samples or datasets) together, we propose to first embed each batch separately in diffusion coordinates, and then find an isometric transformation that aligns these diffusion representations.

In order to find such transformation, we utilize the duality between diffusion coordinates and geometric harmonics that act as generalized Fourier harmonics, as shown in graph signal processing (Shuman et al., 2013). As explained in Section 2.2, this duality allows us to capture cross-batch relations between diffusion coordinates, and orthogonalize the resulting matrix to provide a map between batch-specific diffusion representations. Finally, given an aligned representation, we build a robust unified diffusion geometry that is invariant to both batch effects and batch-specific artifacts. While our approach generalizes naturally to any number of batches, for simplicity, we focus our formulation here on the case of two batches.

## 2.1 DIFFUSION GEOMETRY & GRAPH HARMONICS

The first step in our approach is to capture the intrinsic geometry of each batch $X^{(s)}$ using the diffusion maps method from Coifman & Lafon (2006), which non-linearly embeds the data in a new coordinate system (i.e., diffusion coordinates) that is often considered as representing a data manifold or more generally a diffusion geometry over the data. The diffusion maps construction starts by considering local similarities defined via a kernel $\mathcal{K}(x, y)$, $x, y \in X^{(s)}$ that capture local neighborhoods in the data. We note that a popular choice for $\mathcal{K}$ is the Gaussian kernel $e^{-\frac{\|x-y\|^2}{\sigma}}$, where $\sigma > 0$ is interpreted as a user-configurable neighborhood size. Next, these similarities are normalized to defined transition probabilities $p(x, y) = \frac{\mathcal{K}(x,y)}{\|\mathcal{K}(x,\cdot)\|_1}$ that are organized in an $N \times N$ row stochastic matrix $\mathbf{P}$ that describes a Markovian diffusion process over the intrinsic geometry of the data. Finally, a diffusion map is defined by taking the eigenvalues $1 = \mu_1 \geq \mu_2 \geq \cdots \geq \mu_N$ and (corresponding) eignevectors $\{\phi_j\}_{j=1}^{N}$ of $\mathbf{P}$, and mapping each data point $x \in X^{(s)}$ to an $N$ dimensional vector $\Phi_t(x) = [\mu_1^t \phi_1(x), \ldots, \mu_N^t \phi_N(x)]^T$, where $t$ represents a diffusion-time (i.e., number of transitions considered in the diffusion process). In this work, we denote the diffusion map for the entire dataset $X^{(S)}$ as $\Phi_t^{(s)}$. We refer the reader to Coifman & Lafon (2006) for further details and mathematical derivation, but note that in general, as $t$ increases, most of the eigenvalue weights $\mu_j^t$, $j = 1, \ldots, N$, become numerically negligible, and thus truncated diffusion map coordinates (i.e., using only nonnegligible ones) can be used for dimensionality reduction purposes.

Much work has been done in various fields on applications of diffusion maps as a whole, as well as individual diffusion coordinates (i.e., eigenvectors of $P$), in data analysis (Farbman et al., 2010; Barkan et al., 2013; Mahmoudi & Sapiro, 2009; Angerer et al., 2015; Haghverdi et al., 2016). In particular, as discussed in Coifman & Lafon (2006) and Nadler et al. (2006), the diffusion coordinates are closely related to eigenvectors of Laplace operators on manifolds, as well as their discretizations as eigenvectors of graph Laplacians, which were studied previously, for example, in Belkin & Niyogi (2002). Indeed, the similarities measured in $\mathcal{K}$ can be considered as determining edge weights of a graph structure defined over the data. Formally, we define this graph by considering every data point in $X$ as a vertex on the graph, and then defining weighted edges between them via an $N \times N$ adjacency matrix $\mathbf{W}$ with $\mathbf{W}_{i,j} = \mathcal{K}(x_i, x_j)$, $i, j = 1 \ldots N$. Then, the (normalized) graph Laplacian is defined as $\mathcal{L} = \mathbf{I} - \mathbf{D}^{-1/2} \mathbf{W} \mathbf{D}^{-1/2}$ with $\mathbf{D}$ being a diagonal degree matrix (i.e., with $\mathbf{D}_{i,i} = \sum_{j=1}^{N} \mathbf{W}_{i,j}$). Finally, it is easy to see that $\mathcal{L} = \mathbf{I} - \mathbf{D}^{1/2} \mathbf{P} \mathbf{D}^{-1/2}$, and thus it can be verified that the eigenvectors of $\mathcal{L}$ can be written as $\psi_j = D^{1/2} \phi_j$, with corresponding eigenvalues $\lambda_j = 1 - \mu_j$. It should be noted that if data is uniformly sampled from a manifold (as considered in Belkin & Niyogi, 2002), these two sets of eigenvectors coincide and the diffusion coordinates can be considered as Laplacian eigenvectors (or eigenfunctions, in continuous settings).

A central tenet of graph signal processing is that the Laplacian eigenfunctions $\{\psi_j\}_{j=1}^{N}$ can be regarded as generalized Fourier harmonics (Shuman et al., 2013; 2016), i.e., graph harmonics. In-

---

**Algorithm 1** Manifold Alignment via Feature Correspondence

---

**Input:** Datasets $\mathbb{X} = \{\mathbf{X}^{(1)}, \mathbf{X}^{(2)}\}$ where $\mathbf{X}^{(s)}$ has $N^{(s)}$ observations by $d^{(s)}$ features
**Output:** Aligned graph Laplacian $\mathcal{L}^{(Y)}$.

1: **for** $X^{(s)} \in \mathbb{X}$ **do**
2:     Compute the anisotropic weight matrix $\mathbf{W}^{(s)}$ (Section 2.2) and degree matrix $\mathbf{D}^{(s)}$
3:     Construct the normalized graph Laplacian $\mathcal{L}^{(s)}$ and its truncated eigensystem
$\bar{\mathbf{\Lambda}}^{(s)} = \text{diag}\left[\lambda_i^{(s)}\right]_{i=2}^{N^{(s)}}$, $\bar{\mathbf{\Psi}}^{(s)} = \left[\psi_i^{(s)}\right]_{i=1}^{N^{(s)}}$, with $\mathcal{L}^{(s)}\psi_i^{(s)} = \lambda_i^{(s)}\psi_i^{(s)}$ and $\Lambda \succ 0$
4:     Compute the diffusion map $\mathbf{\Phi}^{(s)} = e^{-t\bar{\Lambda}^{(s)}}\bar{\mathbf{\Psi}}^{(s)}$
5:     The spectral domain wavelet transform tensor $\hat{\mathbf{H}}_{i,j,k}^{(s)}$ ( Equation 1).
6: **end for**
7: Compute intraband harmonic correlations between each dataset $\mathbf{M}'_{:,:,k}$ (Section 2.2).
8: Compute the total interband correlation $\mathbf{M} = \sum_{k=1}^{\tau} \mathbf{M}'_{:,:,k}$.
9: Orthogonalize $\mathbf{M}$ via SVD, $\mathbf{T} = UV^T$
10: Construct the transformed matrix $\mathbf{E} = \begin{bmatrix} \mathbf{\Phi}^{(1)} & e^{-t\bar{\Lambda}^{(1)}}\bar{\mathbf{\Psi}}^{(1)}\mathbf{T} \\ e^{-t\bar{\Lambda}^{(2)}}\bar{\mathbf{\Psi}}^{(2)}\mathbf{T}^T & \mathbf{\Phi}^{(2)} \end{bmatrix}$.
11: Embed $\mathbf{E}$ using a Gaussian kernel to obtain $\mathcal{L}^{(Y)}$.

---

deed, a classic result in spectral graph theory shows that the discrete Fourier basis can be derived as Laplacian eigenvectors of the ring graphs (see, e.g. Olfati-Saber, 2007, Proposition 10). Based on this interpretation, a *graph Fourier transform* (GFT) is defined on graph signals (i.e., functions $f : X^{(s)} \to \mathbb{R}$ over the vertices of the graph) as $\hat{f}(\lambda_j) = \langle f, \psi_j \rangle$, $j = 1, \ldots, N$, similar to the definition of the classic discrete Fourier transform (DFT). Further, we can also write the GFT in terms of the diffusion coordinates as $\hat{f}(\lambda_k) = \langle f, D^{1/2}\phi_j \rangle$, given their relation to graph harmonics. Therefore, up to appropriate weighting, the diffusion coordinates can conceptually be interpreted as intrinsic harmonics of the data, and conversely, the graph harmonics can be considered (conceptually) as intrinsic coordinates of data manifolds. In Section 2.2, we leverage this duality between coordinates and harmonics in order to capture relations between data manifolds of individual batches, and then them in Section 2.3 to align their intrinsic coordinates and construct a unified data manifold over them.

## 2.2 CROSS-GRAPH HARMONIC CORRELATION

We now turn our attention to considering the relation between two batches $X^{(s_1)}, X^{(s_2)}$ via their their intrinsic data manifold structure, as it is captured by diffusion coordinates or, equivalently, graph harmonics. We note that, as discussed extensively in Coifman & Lafon (2006), a naïve construction of an intrinsic data graph with a Gaussian kernel (as described, for simplicity, in Section 2.1) may be severely distorted by density variations in the data. Such distortion would detrimental in our case, as it would the resulting diffusion geometry and its harmonic structure would not longer reflect a stable (i.e., batch-invariant) intrinsic "shape" of the data. Therefore, we follow the normalization suggested in there to separate data geometry from density, and define a graph structure (i.e., adjacency matrix) over each batch via an anistotropic kernel given by

$$\mathbf{W}_{i,j}^{(s)} = \frac{\mathcal{K}(x_i^{(s)}, x_j^{(s)})}{\|\mathcal{K}(x_i^{(s)}, \cdot)\|_1^{1/2} \|\mathcal{K}(x_j^{(s)}, \cdot)\|_1^{1/2}} , i, j = 1, \ldots, N^{(s)}, \ s \in \{s_1, s_2\},$$

where $\mathcal{K}$ is the previously defined Gaussian kernel. This graph structure is then used, as previously described, to construct the intrinsic harmonic structure given by $\{\psi_j^{(s)}\}_{j=1}^N$ on each batch.

While the intrinsic geometry constructed by our graph structures should describe similar "shapes" for the two datasets, there is no guarantee that their computed intrinsic coordinates will match. Indeed, it is easy to see how various permutations of these coordinates can be obtained if some eigenvalues have multiplicities greater than one (i.e., their monotonic order is no longer deterministic), but even beyond that, in practical settings batch effects often result in various misalignments

(e.g., rotations or other affine transformations) between derived intrinsic coordinates. Therefore, to properly recover relations between multiple batches, we aim to quantify relations between their coordinates, or more accurately, between their graph harmonics.

We note that if we even a partial overlap between data points in the two batches, this task would be trivially enabled by taking correlations between these harmonics. However, given that here we assume a setting without such predetermined overlap, we have to rely on other properties that are independent of individual data points. To this end, we now consider the feature functions $\{f_j^{(s)}\}_{j=1}^n$ and our initial assumption that corresponding functions aim to measure equivalent quantities in the batches (or datasets). Therefore, while they may differ in the original raw form, we expect their expression over the intrinsic structure of the data (e.g., as captured by GFT coefficients) to correlate, at least partially, between batches. Therefore, we use this property to compute cross-batch correlations between graph harmonics based on the GFT of corresponding data features. To formulate this, it is convenient to extend the definition of the GFT from functions (or vectors) to matrices, by slight abuse of the inner product notation, as $\hat{\mathbf{X}}^{(s)} = \langle \mathbf{X}, \mathbf{\Psi}^{(s)} \rangle = [\mathbf{\Psi}^{(s)}]^T \mathbf{X}^{(s)}$, where $\mathbf{X}$ consists of data features as columns and $\mathbf{\Psi}^{(s)}$ has graph harmonics as columns (both with rows representing data points).

Notice that the resulting Fourier matrix $\hat{\mathbf{X}}^{(s)}$, for each batch, no longer depends on individual data points, and instead it expresses the graph harmonics in terms of data features. Therefore, we can now use this matrix to formulate a cross-batch harmonic correlations by considering inner products between rows of these matrices. Further, we need not consider all ther correlations between graph harmonics, since we also have access to their corresponding frequency information, expressed via the associated Laplacian eigenvalues $\{\lambda_j^{(s)}\}_{j=1}^{N^{(s)}}$. Therefore, instead of computing correlations between every pair of harmonics across batches, we only consider them within local frequency bands, defined via appropriate graph filters, as exlpained in the following.

Let $g(t)$ be a smooth window defined on the interval $[-0.5, 0.5]$ as $g(t) = \sin\left(0.5\pi \cos(\pi t)^2\right)$. Then, by translating this window along the along the real line, we obtain $\tau$ equally spaced wavelet windows that can be applied to the eigenvalues $\lambda_j^{(s)}$ in order to smoothly partition the spectrum of each data graph. This construction is known as the itersine filter bank, which can be shown to be a tight frame (Perraudin et al., 2014). The resulting windows $g_{\xi_i}(\lambda)$ are centered at frequencies $\Xi = \{\xi_1, \ldots, \xi_\tau\}$. The generating function for these wavelets ensures that each $g_{\xi_i}$ halfway overlaps with $g_{\xi_{i+1}}$. This property implies that there are smooth transitions between the weights of consecutive frequency bands. Furthermore, as a tight frame, this filterbank has the property that $\sum_{i=1}^\tau h_{\xi_i}(\lambda) = 1$ for any eigenvalue. This choice ensures that any filtering we do using the filter bank $G = \{h_{\xi_i}\}_{i=1}^\tau(\lambda)$ will behave uniformly across the spectrum. Together, these two properties imply that cross-batch correlations between harmonics within and between bands across the respective batch spectra will be robust. To obtain such bandlimited correlations we construct the following filterbank tensor

$$\hat{\mathbf{H}}_{i,j,k}^{(s)} = g_{\xi_k}(\lambda_i^{(s)})\psi_i^{(s)T}\mathbf{X}^{(\mathbf{s})}_{:,j} \text{ for } 2 \leq i \leq N^{(s)}. \tag{1}$$

Each $\hat{\mathbf{H}}_{(\cdot,\cdot,k)}^{(s)}$ of this matrix corresponds to the Fourier matrix $\hat{\mathbf{X}}^{(s)}$ with rows scaled by $g_{\xi_k}$. Then, we use these filterbank tensors to compute bandlimited correlations via $\mathbf{M}'_{(\cdot,\cdot,k)} = \hat{\mathbf{H}}_{:,:,k}^{(s_1)}\hat{\mathbf{H}}_{(\cdot,\cdot,k)}^{(s_2)T}$, and finally merge these to generate a combined matrix $\mathbf{M}^{(s_1,s_2)} = \sum_{k=1}^\tau \mathbf{M}'_{(\cdot,\cdot,k)}$, which we refer to as the harmonic (cross-batch) correlation matrix. This step, when combined with the half-overlaps discussed above, allows flexibility in aligning harmonics across bands, which is demonstrated in practice in Section 2.3.

## 2.3 ISOMETRIC ALIGNMENT

Given the harmonic correlation matrix $\mathbf{M}^{(s_1,s_2)}$, we now define an isometric transformation between the intrinsic coordinate systems of the two data manifolds. Such transformation ensures our alignment fits the two coordinate systems together without breaking the rigid structure of each batch, thus preserving their intrinsic structure. To formulate such transformation, we recall that isometric transformations are given by orthogonal matrices, and thus we can rephrase our task as finding the best approximation of $\mathbf{M}^{(s_1,s_2)}$ by an orthogonal matrix. Such approximation is a well studied problem,

dating back to Schönemann (1966), which showed that it can be obtained directly the singular value decomposition $\mathbf{M} = USV^T$ by taking $\mathbf{T}^{(s_1,s_2)} = UV^T$.

Finally, given the isometric transformation defined by $\mathbf{T}^{(s_1,s_2)}$, we can now align of the data manifolds of two batches, and define a unified intrinsic coordinate system for the entire data. While such alignment could equivalently be phrased in terms of diffusion coordinates $\{\phi_j^{(s)}\}_{j=1}^{N^{(s)}}$ or harmonic coordinates $\{\psi_j\}_{j=1}^{N^{(s)}}$, we opt here for the latter, as it relates more directly to the computed harmonic correlations. Therefore, we construct the transformed embedding matrix $\mathbf{E}$ as

$$\mathbf{E} = \begin{bmatrix} \bar{\Psi}^{(s_1)} & [\bar{\Psi}^{(s_1)}]\,\mathbf{T} \\ \bar{\Psi}^{(s_2)}\,\mathbf{T}^T & \bar{\Psi}^{(s_2)} \end{bmatrix} \exp\left( -t \begin{bmatrix} \bar{\Lambda}^{(s_1)} & 0 \\ 0 & \bar{\Lambda}^{(s_2)} \end{bmatrix} \right). \tag{2}$$

where we drop the superscript for $\mathbf{T}$, as they are clear from context, $\bar{\Lambda}^{(s)}$ are diagonal matrices that consists of the nonzero Laplacian eigenvalues of each view, and $\bar{\Psi}$ consist of the corresponding eigenvectors (i.e., harmonics) as its columns. We note that the truncated of zero eigenvalues correspond to zero frequencies (i.e., flat constant harmonics), and therefore they only encode global shifts that we anyway aim to remove in the alignment process. Accordingly, this truncation is also applied to the harmonic correlation matrix $\mathbf{M}^{(s_1,s_2)}$ prior to its orthogonalization. Finally, we note that this construction is equivalent to the diffusion map, albeit using a slightly different derivation of a discretized heat kernel (popular, for example, in graph signal processing works such as (Shuman et al., 2016)), with the parameter $t$ again serving an analogous purpose to diffusion time.

## 3 EMPIRICAL RESULTS

### 3.1 ROTATIONAL ALIGNMENT

As a proof of principle we first demonstrate harmonic alignment of two circular manifolds. To generate these manifolds, we rotated two different MNIST examples of the digit '3' 360 degrees and sampled a point for each degree (See figure 1a). As we noted in section 2.2, the manifold coordinates obtained by diffusion maps are invariant to the phase of the data. In this example it is clear that each '3' manifold is out of phase with the other.

Figure 1b demonstrates the simple rotation that is learned by harmonic alignment between the two embeddings. On the left side, we see the out-of-phase embeddings. Taking nearest neighbors in this space illustrates the disconnection between the embeddings: nearest neighbors are only formed for within-sample points. After alignment, however, we see that the embeddings are in phase with each other because nearest neighbors in the aligned space span both samples and are in the same orientation with each other.

### 3.2 FEATURE CORRUPTION

Next, we assessed the ability of harmonic alignment in recovering k-neighborhoods after random feature corruption (figure 2).

To do this, we drew random samples from MNIST $X^{(1)}$ and $X^{(2)}$ of $N^{(1)} = N^{(2)} = 1000$. Then, for each trial in this experiment we drew $784^2$ samples from a unit normal distribution to create a $784 \times 784$ random matrix. We orthogonalized this matrix to yield the corruption matrix $\mathbf{O}_0$. To vary the amount of feature corruption, we then randomly substituted $\lfloor 0.01 * p * 784 \rfloor$ columns from $\mathbf{I}$ to make $\mathbf{O}_p$. Right multiplication of $X^{(2)}$ by this matrix yielded corrupted images with only $p\%$ preserved pixels (figure 2b, 'corrupted'). To assess the recovery of k-neighborhoods, we then performed a lazy classification on $X^{(2)}\mathbf{O}_p$ by only using the labels of its neighbors in $X^{(1)}$. The results of this experiment, performed for $p = \{0, 5, 10, \ldots 95, 100\}$ are reported in figure 2a. For robustness, at each $p$ we sampled three different non-overlapping pairs $X^{(1)}$ and $X^{(2)}$ and for each pair we sampled three $\mathbf{O}_p$ matrices each with random identity columns, for a total of nine trials per $p$.

In general, unaligned, mutual nearest neighbors (MNN), and harmonic alignment with any filter set cannot recover k-neighborhoods under total corruption; 10% serves as a baseline accuracy that results from the rotation having 10% overlap in the manifold space. On the other hand, for small filter

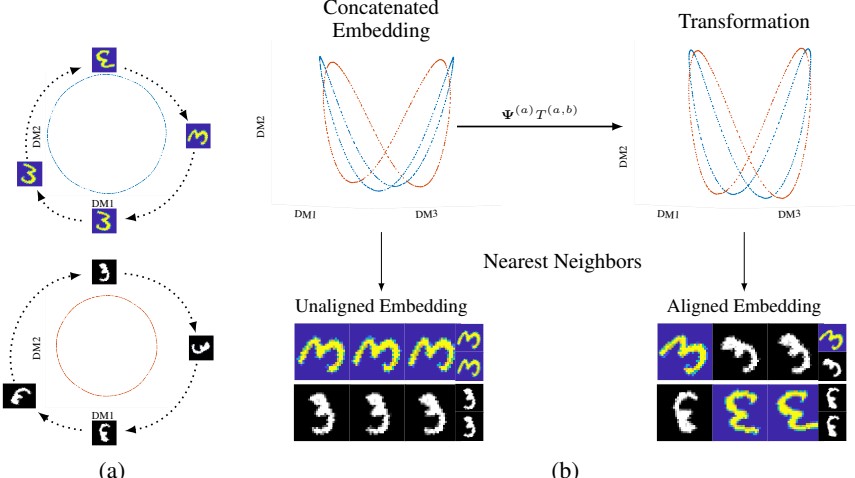

Figure 1: Alignment of circular manifolds. (a) A circular manifold was generated by sampling 360 points from two MNIST examples of the digit '3' rotated in a complete circle. The diffusion geometry that results is circular, but the phase of each digit on the circle is arbitrary. (b) Top: An alignment is obtained by rotating the two embeddings into the latent space of the other. Bottom: The nearest neighbors of a given point in the unaligned embedding are within-sample. Alignment creates connections between samples that are faithful to the phase of the digit, as seen by the presence of out-of-sample nearest neighbors.

choices we observe that harmonic alignment quickly recovers $\geq 80\%$ accuracy and outperforms MNN and unaligned classifications consistently except under high correspondence.

Next we examined the ability of harmonic alignment to reconstruct corrupted data (figure 2b). We performed the same corruption procedure as before with $p = 25$ and selected 10 examples of each digit in MNIST. We show the ground truth from $X^{(2)}$ and the corrupted result $X^{(2)}\mathbf{O}_{25}$ in figure 2b. Then, a reconstruction was performed by setting each pixel in a new image to the dominant class average of the ten nearest $X^{(1)}$ neighbors. In the unaligned case we see that most examples turn into smeared fives or ones; this is likely the intersection formed by $X^{(1)}$ and $X^{(2)}\mathbf{O}_{25}$ that accounts for the 10% baseline accuracy in figure 2a. On the other hand, the reconstructions produced by harmonic alignment resemble their original input examples.

## 3.3 COMPARISONS

In figure 3 we compare the runtime, k-nn accuracy, and transfer learning capabilities of our method with two other contemporary alignment methods. First, we examine the unsupervised algorithm proposed by Wang & Mahadevan (2009) for generating weight matrices between two different samples. The algorithm first creates a local distance matrix of size $k$ around each point and its four nearest neighbors. Then it computes an optimal match between k-nn distance matrices of each pair of points in $X^{(1)}$ and $X^{(2)}$ by comparing all $k!$ permutations of the k-nn matrices and computing the minimal frobenius norm between such permuted matrices. We report runtime results for $k = 5$, as $k = 10$ failed to complete after running for 8 hours. Because the $Wang\&Mahadevan$ (2009) method merely gives a weight matrix that can be used with separate algorithms for computing the final features, we report accuracy results using their implementation. Regardless of input size, we were unable to recover k-neighborhoods for datasets with 35% uncorrupted columns (figure 3b) despite the method's computational cost (figure 3a).

A more scalable approach to manifold alignment has emerged recently in the computational biology (Haghverdi et al., 2018) literature. This approach uses mutual nearest neighbor (MNN) matching to

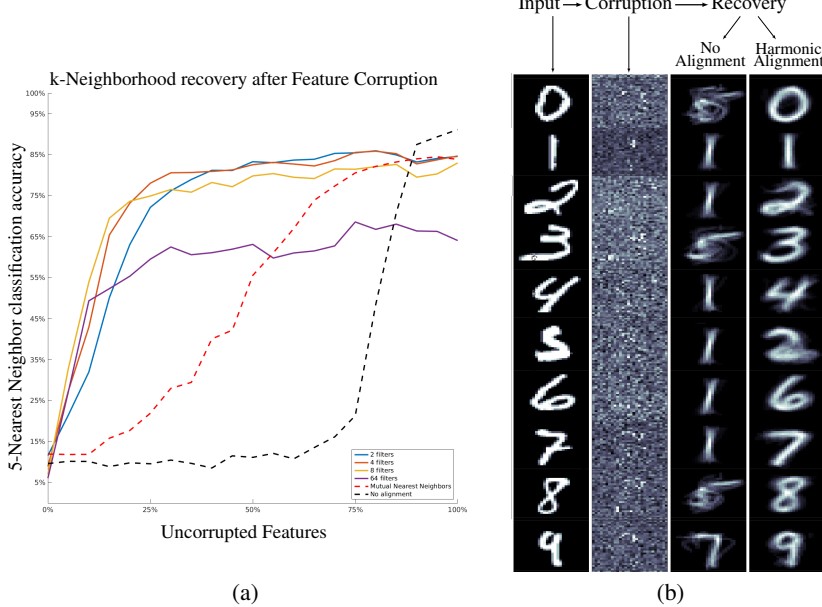

(a)         (b)

Figure 2: Recovery of k-neighborhoods under feature corruption. (a) At each iteration, two sets $X^{(1)}$ and $X^{(2)}$ of $N^{(1)} = N^{(2)} = 1000$ points were sampled from MNIST. $X^{(2)}$ was then distorted by a $784 \times 784$ corruption matrix $\mathbf{O}_p$ for various identity percentages $p$ (see section 3.2). Subsequently, a lazy classification scheme was used to classify points in $X^{(2)}\mathbf{O}_p$ using a nearest neighbor vote from $X^{(1)}$. Results for harmonic alignment with different filterbank sizes, mutual nearest neighbors (MNN), and classification without alignment are shown. (b) Reconstruction of digits with only 25% uncorrupted features. Left: Input digits. Left middle: 75% of the pixels in the input are corrupted. Right middle: Reconstruction without harmonic alignment. Right: Reconstruction after harmonic alignment.

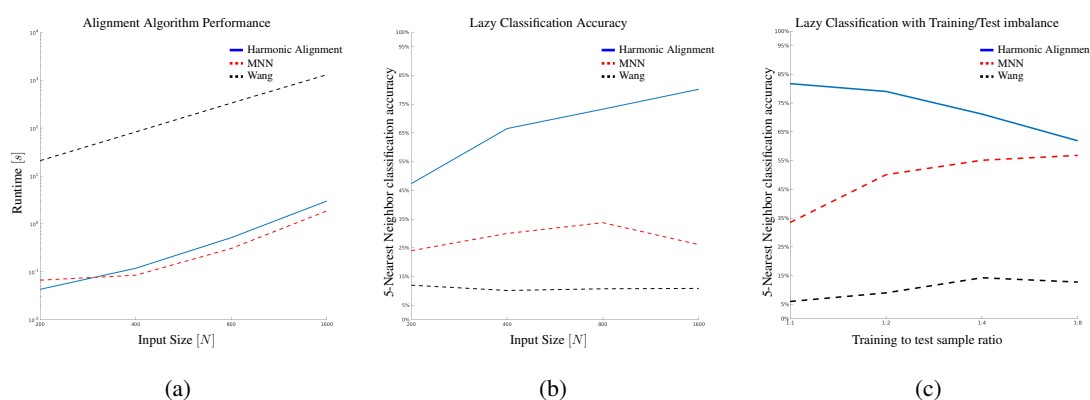

(a)            (b)            (c)

Figure 3: Comparison to other unsupervised alignment methods. (a) Runtime as a function of input size. Algorithhm implementations were obtained from the authors' github repository. Runtime performance was measured on an Intel 3.8 GHz i7-7700HQ laptop with 64 GB Dual-channel DDR4 memory at 2400 MHz running Pop!OS 4.15 and MATLAB R2018a. (b) Lazy classification accuracy relative to input size. For each input size $N$, the average of 3 iterations of lazy classification of $N/2$ unlabeled randomly corrupted digits with 35% preserved pixels (see section 3.2) is reported. (c) Transfer learning performance. For each ratio, 1,000 uncorrupted, labeled digits were sampled from MNIST. 1,000, 2,000, 4,000, and 8,000 (x-axis) unlabeled points were sampled and corrupted with 35% column identity. The mean of three iterations of lazy classification for each method is reported.

compute mapping between datasets based on the assumption that if two points are truly neighbors they will resemble each other in both datasets. Because this approach amounts to building a k-nearest neighbors graph for each dataset and then choosing a set of neighbors between each dataset, MNN scales comparably to our method (figure 3a. Additionally, MNN is able to recover 20-30% of k-neighborhoods when only 35% of features match (figure 3b); this is an improvement over Wang but is substantially lower than what harmonic alignment achieves. We note that the performance of harmonic alignment was correlated with input size whereas MNN did not improve with more points.

## 3.4 TRANSFER LEARNING

An interesting use of manifold alignment algorithms is transfer learning. In this setting, an algorithm is trained to perform well on one dataset, and the goal is to extend the algorithm to the other dataset after alignment. In figure 3c we explore the utility of harmonic alignment in transfer learning and compare it to MNN and the method proposed by Wang & Mahadevan (2009).

In this experiment, we first draw 1000 uncorrupted examples of MNIST digits, and construct a diffusion map to use as our training set. Next, we took 65% corrupted unlabeled points (see section 3.2) in batches of $1,000, 2,000, 4,000, 8,000$ as a test set to perform lazy classification on using the labels from the uncorrupted examples. At 1:8 test:training sample sizes, Harmonic alignment outperformed Wang and MNN by aligning upto 60% correct k-neighborhoods. In addition to showing the use of manifold alignment in transfer learning, this demonstrates the robustness of our algorithm to dataset imbalances.

## 3.5 BIOLOGICAL DATA

To illustrate the need for robust manifold alignment in computational biology, we turn to a simple real-world example obtained from Amodio et al. (2018) (figure 4). This dataset was collected by mass cytometry (CyTOF) of peripheral blood mononuclear cells (PBMC) from patients who contracted dengue fever. Subsequently, the Montgomery lab at Yale University experimentally introduced these PBMCs to Zika virus strains.

The canonical response to dengue infection is upregulation of interferon gamma (IFN$\gamma$) (Chesler & Reiss, 2002; Chakravarti & Kumaria, 2006; Braga et al., 2001). During early immune response, IFN$\gamma$ works in tandem with acute phase cytokines such as tumor necrosis factor $\alpha$ to induce febrile response and inhibit viral replication (Ohmori et al., 1997). In the PBMC dataset, we thus expect to see upregulation of these two cytokines together, which we explore in 4.

In figure 4a, we show the relationship between IFN$\gamma$ and TNF$\alpha$ without denoising. Note that there is a substantial difference between the IFN$\gamma$ distributions of sample 1 and sample 2 (Earth Mover's Distance (EMD) = 2.699). In order to identify meaningful relationships in CyTOF data, it is common to denoise it first. We used a graph filter to denoise the cytokine data (Van Dijk et al., 2018). The results of this denoising are shown in figure 4b. This procedure introduced more technical artifacts by enhancing the difference between batches, as seen by the increased EMD (3.127) between the IFN$\gamma$ distributions of both patients. This is likely due to a substantial connectivity difference between the two batch subgraphs in the total graph.

Next, we performed harmonic alignment of the two patient profiles. We show the results of this in figure 4c. Harmonic alignment corrected the difference between IFN$\gamma$ distributions and restored the canonical correlation of IFN$\gamma$ and TNF$\alpha$. This example illustrates the utlity of harmonic alignment for biological data, where it can be used for integrated analysis of data collected across different experiments, patients, and time points.

## 4 CONCLUSION

We presented a novel method for aligning or batch-normalizing two datasets that involves learning and aligning their intrinsic manifold dimensions. Our method leverages the fact that common or corresponding features in the two datasets should have similar harmonics on the graph of the data. Our *harmonic alignment* method finds an isometric transformation that maximizes the similarity of frequency harmonics of common features. Results show that our method successfully aligns artifi-

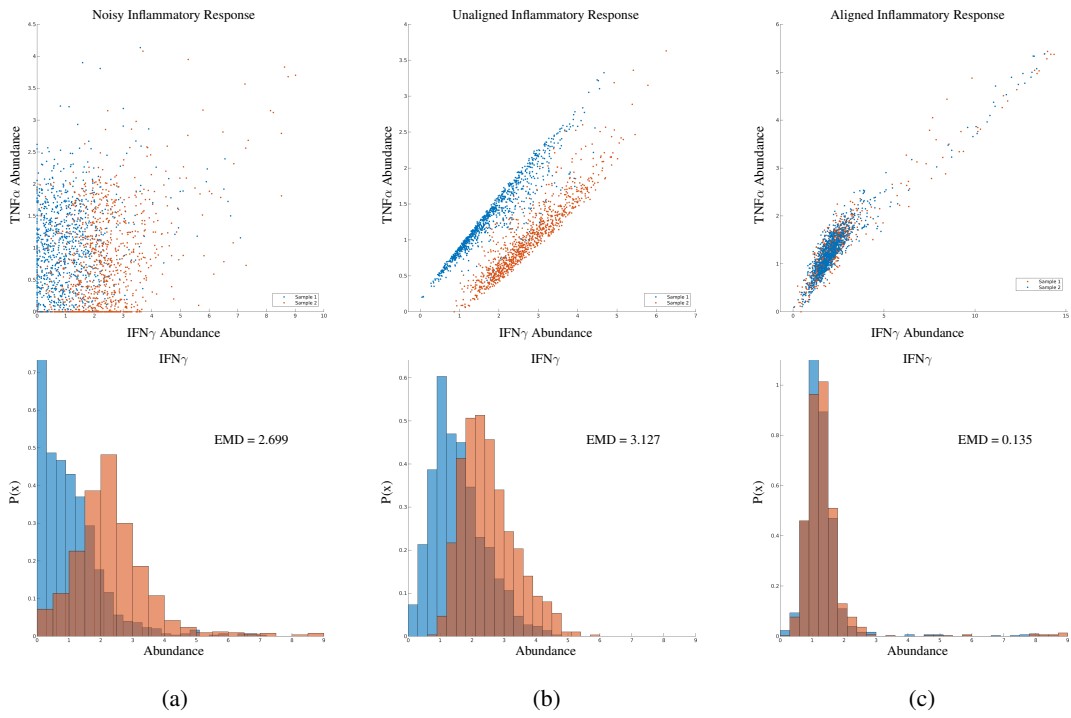

Figure 4: Batch effect removal in biological data. 4,000 cells were subsampled from two single-cell immune profiles obtained via mass cytometry on blood samples of two patients infected with Dengue fever. Top: Both patients exhibit heightened interferon gamma (x-axis), a pro-inflammatory cytokine which is associated with tumor necrosis factor alpha (TNF$\alpha$, y-axis) Bottom: IFN$\gamma$ histograms for each batch. EMD: "Earth Mover's Distance" (a) Data before denoising. (b) Denoising of unaligned data enhances a technical effect between samples in IFN$\gamma$. (c) Harmonic alignment corrects the IFN$\gamma$ shift.

cially misaligned as well as biological data containing batch effect. Our method has the advantages that it aligns manifold geometry and not density (and thus is insensitive to sampling differences in data) and further our method denoises the datasets to obtain alignments of significant manifold dimensions rather than noise. Future applications of harmonic alignment can include integration of data from different measurement types performed on the same system, where features have known correlations.

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
