# OpenReview forum: "Manifold Alignment via Feature Correspondence"
_ICLR.cc/2019/Conference_

### Official Review · AnonReviewer2 · 2018-10-29
**Manifold Alignment via Feature correspondence - Review**

**Rating:** 4
**Confidence:** 3

**Review:**

The paper proposes an alignment of two manifolds that is performed in a low-dimensional parameter space corresponding to a low-pass "filtering" of the Graph Fourier transform obtained from the underlying data graphs for the two manifolds. The numerical results show the quality of the alignment for some toy image datasets and a biological dataset.

The derivation of the technical details of the approach is not clear - see the comments below on Pages 5,6 and 9 in particular. The paper is not clear enough for acceptance at this point.

Detailed comments:

Page 2: Grammar error "that is invariant batch effects". When denoising is discussed, can you explain whether this is denoising or simply regularization? When is the selected subspace a good approximation for the "signal subspace"?
Page 3: Should X^(S) be X^(s)? When W and W(s) are defined, do they also rely on a neighborhood graph? It appears that in the definition of psi_j the eigenvectors phi_j should be obtained from W, not P (which is how they are defined earlier in the page).
Page 4: There is an abuse of notation on f, used both as a linear function on X(s) and an element of X(s).
Page 5: Typos "exlpained", "along the along the". It is not clear what applying a window to eigenvalues means, or what the notation g_xi(lambda) means. The construction of the filters described here needs to be more explicit. h_xi is undefined. How is H in (1) defined when i = 1?
Page 6: M should be M(s1,s2). Typesetting error in Lambdabar(s). Which matrix is referred to in "the laplacian eigenvalues of each view"? What is the source and target of the embedding E? How is the embedding applied to data x(s1), x(s2)?
Page 7: Figure 1a appears to have an error in the orientation of one of the blue "3"s. The text on the arrow between the manifold embeddings does not agree with the notation in the paper. In Figure 1b, it is not clear which image is the original point and which images are the neighbors, or why some images are smaller than others. Results for the other algorithms are missing (why no comparison?). Typo "Wang&Mahadevan". Can you be more specific as to why that algorithm was "unable to recover k-neighborhoods" in certain cases?
Page 8: Why no comparison with Wang & Mahadevan in Figure 2?
Page 9: There is little description as to how manifold learning is applied in the biological data example. What is the ambient dimensionality and the dimension of the manifolds? How are the "abundances" extracted from the data?
"Which we explore in 4" -> "Which we explore in Fig. 4"

---

### Official Review · AnonReviewer1 · 2018-11-02
**Could benefit from a revision including more comprehensive experiments.**

**Rating:** 5
**Confidence:** 4

**Review:**

This paper tackles the problem of noisy measurements collected from same samples that may result in different experimental data collection scenarios, especially in biology. Given S different data batches of the same samples, this paper aims to perform manifold alignment between each batch using feature correspondences.
The motivation is that even though data points from each experiments may differ due to noise coming from different factors, they should at least exhibit some correlations in the feature space. Such correlation is exploited by embedding each batch of data in a space represented by diffusion coordinates computed using an anisotropic kernel. Inter-batch correlations are computed between diffusion coordinates of each batch, which are exploited to construct an isometric transformation between each pair of diffusion coordinates of batch datasets. The later transformation is used to construct an aligned graph Laplacian where each batch have similar representations.

This paper tackles an important problem using a novel approach where instead of aligning each pair of data-points it is attempting to align geometries of batch specific manifolds. The authors show through a toy experiment on MNIST that the proposed algorithm indeed is able to align manifolds accurately. Moreover it is also able to perform manifold denoising and achieves superior classification results compared to two existing approaches. Finally, the proposed algorithm is applied to a practical biological case and shows that it is indeed able to align data from two different immune profiles.

Although the paper tackles efficiently an important problem, I am concerned about the experimental section and think it would be improved by taking the following points into account:

•	The proposed approach is compared only with two other algorithms. For example, one could compare the denoising ability to [Hein and Maier 2006: manifold denoising] or [Cui et al: Generalized unsupervised Manifold Alignment] for manifold alignment. Furthermore, it would be informative to see how the proposed approach compares to recent domain adaptation approaches as they attempt to map data from different domains into a shared
representation which is rather similar to what the proposed algorithm is doing.
•	Experiments are all performed using rather simple datasets. It would be interesting to see how the algorithms would perform on slightly more complicated images such as Cifar 10 for example.
•	It is not clear what are the next steps to perform after obtaining Eq. 2
•	It is not clear what are the number of filters in Figure 2 a).
•	Figure 1 needs to be clarified further: What are DM1, DM2, DM3 mean? What are the columns and rows in Figure1-b (bottom)?

---

### Official Review · AnonReviewer3 · 2018-11-04
**An Interesting and Straightforward Proposal on Variation Alignment but the Argument and Evidence is Not Strong Enough.**

**Rating:** 5
**Confidence:** 4

**Review:**

The authors pointed out that the measurements in biology and natural science suffer from batch effects such as the variations between batches of data measured at different times or by different sensors. In order to analyze different batches of data, an alignment or a calibration is frequently needed. The authors propose to use that though there is variation among different batches, these batches all share an underlying intrinsic manifold structure, which may admit a set of alignable coordinates.

Technically, the authors propose to choose the diffusion kernel method, which is one of the spectral methods, to extract the harmonic like eigenfunctions defined on the manifold, for each of the batches of data. Using these harmonic-like coordinates, the authors assume there exists an isometric rotation in the between each pair of batches such that their coordinates can be aligned under this orthogonal rotation.

Comments:
Overall I think the problems pointed out do exist and this is an interesting proposal to use the manifold structure to align the data. But there are some weak points in this proposal:
1. It's well known that spectral methods are frequently sensitive to perturbations of the datasets. At the beginning of section 2.2 the authors propose to use a normalization to construct the kernel, however, I don't quite understand how this would solve the instability to perturbations.

2. In my opinion, the equation (1) is the most interesting construction in this paper. This motivation for this tensors construction is not strong enough and I would suggest put more detail into this construction. My understanding is the window functions g introduced here serve for an invariance purpose such that when the frequency slightly shift (or rotate), the correlation computed should be stable. But the tradeoff of choosing a proper window should be discussed carefully, potentially with different dataset since different dataset may have a different sensitivity to perturbations across different batches.

3. In section 3.1, the first motivating example is quite confusing. The authors demonstrated the alignment of two rotated MNIST digits, 3. For each digit, the underlying manifold is S1. S1 is diffeomorphic to its rotation. So I'm not so sure what's the underlying manifold geometry used to align them.  My understanding is that this alignment doesn't come from the S1 manifold but comes from some additional structure in the image signal. Fig1(b) is also a little confusing that I couldn't figure out what's drawn there.

Overall, to use the underlying manifold structure to align data batches is an interesting and straightforward proposal, but I hope the authors can address these question carefully and make the argument stronger.

---

### Meta-Review · Area_Chair1 · 2018-12-17
**empirical analysis somewhat simplistic with inadequate comparisons to other correspondence construction methods**

**Confidence:** 5
**Recommendation:** Reject

**Metareview:**

The diffusion maps framework is used to embed a given collection of datasets into diffusion coordinates that capture intrinsic geometry. Then a correspondence map is constructed between datasets by  finding rotations that align these coordinates. The approach is interesting. The reviewers, however, found the empirical analysis somewhat simplistic with inadequate comparisons to other correspondence construction methods in the literature.